# Membrane Fouling Controlled by Adjustment of Biological Treatment Parameters in Step-Aerating MBR

**DOI:** 10.3390/membranes11080553

**Published:** 2021-07-22

**Authors:** Dimitra C. Banti, Manassis Mitrakas, Petros Samaras

**Affiliations:** 1Laboratory of Technologies of Environmental Protection and Utilization of Food By-Products, Department of Food Science and Technology, International Hellenic University, GR-57400 Thessaloniki, Greece; samaras@ihu.gr; 2Laboratory of Analytical Chemistry, Department of Chemical Engineering, Aristotle University of Thessaloniki, GR-54124 Thessaloniki, Greece; mmitraka@cheng.auth.gr

**Keywords:** wastewater treatment, filamentous microorganisms, membrane bioreactor, membrane fouling, operating conditions

## Abstract

A promising solution for membrane fouling reduction in membrane bioreactors (MBRs) could be the adjustment of operating parameters of the MBR, such as hydraulic retention time (HRT), food/microorganisms (F/M) loading and dissolved oxygen (DO) concentration, aiming to modify the sludge morphology to the direction of improvement of the membrane filtration. In this work, these parameters were investigated in a step-aerating pilot MBR that treated municipal wastewater, in order to control the filamentous population. When F/M loading in the first aeration tank (AT_1_) was ≤0.65 ± 0.2 g COD/g MLSS/d at 20 ± 3 °C, DO = 2.5 ± 0.1 mg/L and HRT = 1.6 h, the filamentous bacteria were controlled effectively at a moderate filament index of 1.5–3. The moderate population of filamentous bacteria improved the membrane performance, leading to low transmembrane pressure (TMP) at values ≤ 2 kPa for a great period, while at the control MBR the TMP gradually increased reaching 14 kPa. Soluble microbial products (SMP), were also maintained at low concentrations, contributing additionally to the reduction of ΤΜP. Finally, the step-aerating MBR process and the selected imposed operating conditions of HRT, F/M and DO improved the MBR performance in terms of fouling control, facilitating its future wider application.

## 1. Introduction

Membrane bioreactors (MBRs) have been widely used during the last decades for wastewater treatment worldwide in large-scale (≥10,000 m^3^/d) or even in super-large-scale (≥100,000 m^3^/d) plants, due to their various advantages, such as excellent effluent quality [1], occupation of small land area, low hydraulic retention time (HRT), higher solids retention time (SRT) and lower sludge production [2,3,4]. However, they present a major disadvantage that resides in energy consumption, which is attributed to the membrane fouling problem [5,6,7]. Membrane fouling is caused by the accumulation and deposition of activated sludge substances on the surface and in the pores of the membrane [8,9,10,11,12]. According to a recent research work [13], the combination of colloids and SMP (Soluble Microbial Products) -of protein-like and polysaccharide-like substances [14]- of activated sludge mixed liquor are the main parameters that worsen membrane fouling, while colloids play an even more decisive role. Membrane fouling leads to an increase in trans-membrane pressure (TMP) by reducing membrane permeability [3].

Extensive studies have been performed aiming to predict membrane fouling [15], to explain the membrane fouling mechanism [16,17,18,19], as well as to confront it [20,21,22]. According to such research works, gradual deposition and aggregation of SMP in the pores of the membrane caused irreversible membrane fouling, having, as a result, the pore blockage and increase of fouling rate and/or the detachment of aggregates in the permeate [16]. Moreover, according to past research activity [23,24], the filamentous bacterial population, known until today for causing bulking sludge problems in wastewater treatment processes, was manipulated through a modified MBR configuration and achieved to confront effectively the membrane fouling problem. However, this work needs to be optimized by further research.

According to some research works [25,26], a high Food/Microorganisms (F/M) ratio or Organic Loading Rate (OLR) induced the generation of SMP and more bound extracellular polymeric substances (EPS), resulting in a decrease of sludge filterability and lower filtration index. On the other hand, low Dissolved Oxygen (DO) increased filaments concentration and especially *Type 0041* and *M. parvicella* [27,28]. Regarding the favorable DO concentration, it has been reported that the limitation or deficiency of DO was often responsible for the proliferation of filamentous bacteria in activated sludge processes [29,30]. However, on the other hand, [31] also reported that an increase in the DO levels from 1–2.5 mg/L to 3.5–5.7 mg/L under thermophilic conditions (55 °C) led to an increase in the filaments level.

Furthermore, a decrease of HRT from 24 h to 18 h [32], resulted in the release of EPS from the bacterial cells, which was responsible for the rise in SMP and sludge deflocculation. Nevertheless, according to the authors SMP rise was also attributed to other unstable operating conditions and not necessarily to the decreased HRT. Reduction of HRT from 12 h to 5 h [26], also, caused the growth of filamentous bacteria and the formation of large and irregular flocs. However, too high HRT also led to the accumulation of foulants [26,32]. Short SRT of 10 d [33] or 23 d compared to high SRT of 40 d [34] harmed membrane fouling due to high concentrations of SMP and EPS, mainly in the form of polysaccharides [35]. At shorter SRT, the concentrations of deposited EPS on the membrane were much higher in comparison to higher SRT. Although prolonged SRT minimized excess biomass production, too long SRT (>60 d) accelerated fouling due to large amounts of foulants and high sludge viscosity [34,36].

Regarding the correlation of filamentous bacteria to SMP, EPS and sludge characteristics, it has been reported that filamentous bacteria density had no significant effect on bound extracellular polymeric substances (bound EPS, r_p_ =−0.343, *p* = 0.080) and soluble microbial products (SMP, r_p_ = 0.221, *p* = 0.267) [37]. However, they had an important impact on floc size (r_p_ = 0.944, *p* = 0.000) and floc structure (r_p_ = −0.752, *p* = 0.000). A high filamentous index contributed to the formation of larger flocs loosely structured and vice versa. Therefore, according to this research work, although filamentous bacteria can change the floc morphology, their effect on the membrane-fouling rate might be negligible. However, more recent work has discovered that EPS content and components were changed during filamentous bulking [38]. Especially, during sludge bulking, proteins and polysaccharides of EPS were gradually decreased, sludge hydrophobicity reduced, and surface negative charge increased. However, no information has been given about the fate of the SMP in bulking sludge conditions.

Considering the operating parameters of the MBRs, which affect the membrane foulants, significant research work has been carried out on MBR process to reduce membrane fouling, including contradictory conclusions or shortcomings. Previous work identified the crucial role of filamentous bacteria for membrane fouling control [23,24]. However, as filamentous bacteria present a significantly lower adsorption rate of soluble components, at the typical wastewater treatment processes their population is diminished due to the faster adsorption rate of the other floc-forming bacteria. The step-aeration process focuses on the modification of biological treatment parameters to decrease floc-forming bacteria absorption rate, favoring, in turn, the development of filamentous bacteria. Thus, this work aims to optimize operating conditions towards minimization of fouling, using controlled filamentous population, by adjusting the F/M loading, HRT and DO in the two aerated tanks of a step-aerating MBR. Moreover, the effect of filamentous bacteria on activated sludge characteristics and sludge metabolism products, such as SMP, which are the key-foulants of the filtration membranes, was investigated to achieve the membrane fouling reduction. The step-aerating MBR process that is proposed by this work to control filamentous population, is a clean and cost-effective solution to reduce membrane fouling in MBRs. The step-aerating MBR reduces the operating cost of the MBR, as the frequency of the chemical cleanings and replacements of the fouled membranes are minimized. Moreover, this advantageous process for MBRs is not based on any chemical addition—a solution highly recommended by researchers nowadays [39,40]—but only on the modification of operational conditions, offering therefore an additional economic and environmental benefit. By the proposed methodology of control of the filamentous population, the bulking sludge problem, which is usually attributed to the overgrowth of filamentous microorganisms, is also solved, and therefore a beneficial solution could be given for conventional wastewater treatment units too.

## 2. Materials and Methods

### 2.1. Step-Aerating and Control MBR Set-Up

The step-aerating pilot-scale membrane bioreactor presented in Figure 1, was consisted of two in-series aerated tanks: the first aeration tank (AT_1_) of working volume 5 L and the second aeration tank (AT_2_) of working volume 15 L. The dissolved oxygen in both aerated tanks was regulated at 2.5 ± 0.1 mg/L. A hydrophilic flat sheet microfiltration (MF) membrane module, further described in Appendix A, was submersed in the membrane tank (MT). Intense aeration was provided to the bottom of the MT to scour the cake layer from the membrane’s surface with an air sparging rate of 10 L/min (5.5 m^3^/m^2^·h). The filtration was carried out by a filtration step of 10 min duration followed by a pause step of 2 min. The MBR was located in a closed and protected tank, where the temperature was controlled at 20 ± 3 °C in order not to affect the process.

The MBR contained DO meters, an air compressor (CO), a thermometer and three peristaltic pumps, from which P_1_ was the feed pump of the influent synthetic wastewater (Q_in_ = 0.9 L/h), P_2_ was the recirculation pump (Q_r_ = 2.2 L/h) and P_3_ was the effluent pump (Q_eff_ = 1.1 L/h). The operating parameters, such as TMP, DO, P_1_, P_2_, P_3_ and temperature (T) were recorded and controlled by a Programmable Logic Controller (PLC, Eutech Instruments, Singapore) using SCADA software (Simantec, Siemens (Munich, Germany), Version 14).

Synthetic wastewater was prepared two times per week to feed the MBR and contained glucose, corn starch, NH_4_Cl, peptone, KH_2_PO_4_, MgSO_4_·7H_2_O, MnSO_4_·H_2_O and FeSO_4_·7H_2_O in the concentrations described further in [13]. pH was controlled at 7.0–7.5, adding NaHCO_3_. The chemical oxygen demand (COD) value of synthetic wastewater was ranged at 890 ± 120 mg/L. At the beginning of the MBR operation, the tanks were filled with activated sludge mixed liquor taken from a full-scale wastewater treatment plant of central Macedonia in Greece. An adjustment period of about 10 d was used to acclimate synthetic wastewater to the activated sludge process. Throughout the duration of the experiment only one membrane was used, which was never cleaned or replaced, as no total membrane fouling was observed.

Mixed liquor suspended solids (MLSS) in the step-aerating MBR were controlled at 6000 ± 1500 mg/L. The food/microorganisms (F/M) loading was, also, controlled at 0.65 ± 0.20 g COD/g MLSS/d in the AT_1_ and 0.03 ± 0.01 g COD/g MLSS/d in the AT_2_. The F/M ratio in the total MBR unit was equal to 0.13 ± 0.04 g COD/g MLSS/d.

For comparison needs, a control MBR was operated in parallel with the step-aerating MBR, which had similar operating conditions, with, however, a basic difference, that it consisted of a single aerobic chamber of 20 L working volume, followed by a membrane tank of 5 L working volume. In more detail, the membrane tank contained a similar submerged flat sheet microfiltration membrane module with the step-aerating MBR. The control MBR was constantly fed with synthetic wastewater of the same composition as the step-aerating MBR, having an influent COD equal to 850 ± 76 mg/L. MLSS concentration was equal to 6000 ± 1000 mg/L, while dissolved oxygen concentration was controlled at the typical value of 2.5 ± 0.1 mg/L. The temperature was, also, controlled at 20 ± 3 °C in order not to affect the process. The F/M ratio in the control MBR was 0.12 ± 0.03 g COD/g MLSS/d. Since the control MBR operated only for comparative reasons, the only parameters that were measured were TMP, COD, MLSS and Filamentous Index. A comparable summary table of the operating conditions of the step-aerating MBR and control MBR is presented in Table 1, where many parameters are described in the form of average ± standard deviation.

### 2.2. Determination of Physicochemical Parameters

COD (Chemical oxygen demand), N-NH_4_, N-NO_3_ and total N were measured for influent synthetic wastewater as well as the permeate using Hack–Lange LCK kits and a DR-2800 spectrophotometer for a sample per week. Furthermore, MLSS were measured by applying standard methods [41] for a sample per week.

### 2.3. Critical Flux Determination and Stabilization of the Reduction Tendency of the Permeate Flux

Aiming to control membrane fouling and maintain a sustainable operation, critical flux was measured for both MBR units, defining the flux below which mainly reversible fouling occurs [42,43]. Therefore, the critical flux of each MBR was evaluated using the flux-step method [42,43] and it was found to be slightly greater than 10.2 Lmh for both of them. Therefore, the MBRs were adjusted to operate in subcritical flux conditions, at 10 Lmh and thus the influent (Q_in_), recirculation (Q_r_) and effluent flowrates (Q_eff_) was adjusted at 0.9, 2.2 and 1.1 L/h, respectively. The Q_eff_ remained constant throughout the whole duration of the experiment. The difference in values between Q_in_ and Q_eff_ was 0.2 L/h as the permeate flux was intermittent with an operation step of 10 min followed by a pause step of 2 min. Therefore, the flows were properly adjusted to have an equilibrium in the MBR unit.

Directing to keep flux Q_eff_ constant, despite the increasing pumping pressure (TMP) in the membrane as a result of the gradual membrane fouling, a tank having level gauges was added at the end of the unit (Figure 1). In this tank, the flow rate was measured and when the flow was decreased (due to the increased pumping pressure in the membrane), the peristaltic pump P_3_ was commanded through PLC to increase its speed.

### 2.4. Determination of Soluble Microbial Products (SMP)

SMP were extracted using a physical extraction method as described by [16,44], while the SMP extracts were further analyzed to determine their protein and carbohydrate content. The Dubois photometric method was applied to measure their content in carbohydrates [45], in duplicate for each measurement, whereas a modified Lowry method was used to measure protein concentration [46], in triplicate for each sample, because of the sensitivity of the measurement. Bovine serum albumin (BSA, Sigma Aldrich) and glucose (Panreac) were used to calibrate protein and carbohydrate measurement respectively.

### 2.5. Filamentous Index (FI) Measurement and Characterisation of Filamentous Microorganisms

Filamentous Index (FI) was used, according to the Eikelboom method [27,28], to measure the population of filamentous microorganisms in the activated sludge mixed liquor of the MBR units. For the FI measurement, a Light Sheet Microscope (LSM, Observer Z1, Zeiss, Oberkochen, Germany) was used mainly in 100× magnification, whereas in some cases 50× and 200× magnifications were, also, used to take a better view of the samples. In filamentous index measurement, FI = 0 corresponds to no filaments coming out of the sludge flocs, whereas FI = 5 corresponds to infinite filaments. ZEN software for microscope and imaging was used to edit the images, creating tiff image files. FI was measured for mixed liquor samples from all the tanks of both MBR units and they were found similar for the tanks of each MBR unit, as both MBRs were in balance. Gram staining and Neisser staining procedures [28] were, also, performed using the Light Sheet Microscope (LSM, Observer Z1, Zeiss) in 200× magnification to identify filamentous microorganisms.

## 3. Results

### 3.1. Adjustment of Operating Conditions in the Step-Aerating MBR Unit

The operating conditions at the AT_1_ of the step-aerating MBR were controlled at a high F/M loading of 0.65 ± 0.2 g COD/g MLSS/d and a typical DO concentration of 2.5 ± 0.1 mg/L, whereas, at the AT_2_, too low F/M of 0.03 ± 0.01 g COD/g MLSS/d and a typical DO of 2.5 ± 0.1 mg/L was supplied. DO values were maintained at 2.5 ± 0.1 mg/L for about 3 months (81 days) in both AT_1_ and AT_2_ chambers aiming to facilitate the biological treatment process and this stage of 81 days is called hereafter Stage 1. Following, DO was decreased in the AT_1_ at 1.2 ± 0.5 mg/L aiming to study its effect on filamentous growth, SMP concentration and membrane fouling, while DO at the AT_2_ was kept constant at 2.5 ± 0.4 mg/L throughout the experiment (Figure 2). The stage between the 82nd day and until the total membrane fouling will be called hereafter Stage 2.

The HRT based on Q_in_ + Q_r_ was 1.6 h in the AT_1_ tank, 4.8 h in the AT_2_, while the corresponding HRT based only on Q_in_ was equal to 5.5 h, 17 h in the AT_1_ and AT_2_ respectively, and 28 h for the whole MBR unit. The sludge retention time was 30 ± 5 d. The operating parameters in the step-aerating MBR unit compared to the control MBR unit are summarized in Table 1. The HRT values were intentionally given to the MBR unit aiming to keep the membrane fouling rate as low as possible.

### 3.2. Transmembrane Pressure (TMP) and Membrane Fouling Profiles

The TMP profile as a function of the MBR operation time for the step-aerating MBR is presented in Figure 3. According to the blue line of Figure 3, rotations of the P_3_ pump were gradually automatically increased, aiming to correct the operation of the peristaltic pump and to keep Q_eff_ constant and equal to 1.1 L/h even when membrane fouling was increased. Figure 3 also represents the TMP evolution of the control MBR as a function of operation time.

### 3.3. Growth, Control and characTerisation of Filamentous Microorganisms

The filament index in the step-aerating MBR is presented in detail in Figure 4. Following, Figure 5 presents optical microscopy images, where the gradual growth of filamentous microorganisms is presented. Regarding the control MBR, the filamentous population was retained at very low levels of FI < 1.5.

The day when the maximum membrane fouling was observed in the MBR unit that was the 121 day of MBR operation, the filamentous microorganisms were stained and characterized, aiming to determine the reason for their growth. The greatest population of filamentous microorganisms according to Figure 6a,b, was Gram-negative and Neisser positive and they were named as *Type 0092* filaments. According to Figure 6c,d, *Thiothrix* spp. filaments were, also, detected having a yellow/brown color after Neisser staining. Finally, Figure 6e,f illustrate that the filaments *Microthrix parvicella* were, also, present in the mixed liquor.

### 3.4. SMP Carbohydrates and Proteins Concentrations in the Step-Aerating MBR

SMP carbohydrates were maintained as low as 8 ± 6 mg/L in the step-aerating MBR, as it is presented in Figure 7a, in activated sludge samples coming from the AT_2_ tank, while in general lines ranged at concentrations smaller than 15 mg/L. A gradual increase of SMP carbohydrates after the first 30 days was attributed to a corresponding gradual increase of MLSS. However, the conclusion was not affected by this incident as their total concentration remained smaller than 15 mg/L. SMP carbohydrates were, also, measured for the AT_1_ and MT tanks and found similar to the AT_2_’s values, as the MBR unit was in balance. SMP carbohydrates in the permeate were found equal to 5 ± 3 mg/L, which implied that a great part of the SMP passed through the membrane.

Respectively, SMP protein concentrations were maintained as low as 11 ± 6 mg/L (Figure 7b) for activated sludge samples from the AT_2_, and generally, their concentrations ranged at values below 20 mg/L. Similar SMP protein concentrations were found for the AT_1_ and MT tanks. SMP proteins in the filtrate were found to be equal to 7 ± 4 mg/L.

### 3.5. Wastewater Treatment Efficiency

Wastewater treatment efficiency is presented in Table 2 in the form of average ± standard deviation, regarding chemical oxygen demand (COD), nitrate (NO_3_-N), ammonium (NH_4_-N) and total nitrogen (TN) concentrations for the influential wastewater and the effluent permeate. As it is observed, COD removal was excellent, reaching 98%, verifying that COD removal was not affected by the increase of filamentous microorganisms. NH_4_-N content was decreased at the effluent and therefore nitrification was efficient. However, NO_3_-N was significantly increased at the effluent because of the absence of an anoxic denitrification step in the wastewater treatment process. TN removal was particularly low, equal to 24%, and it could be attributed to the incorporation into biomass synthesis alone.

Wastewater treatment regarding the reduction of biological loading in the control MBR was highly effective, as the outflow COD was found equal to 13 ± 4.2 mg/L. Other physicochemical parameters were not measured in the control MBR as the purpose of its operation was just the comparative study of membrane fouling compared to the step-aerating MBR.

## 4. Discussion

As it can be seen in Figure 3, TMP at Stage 1 was kept under constant control at values lower than 2 kPa. Therefore, the significantly high F/M = 0.65 ± 0.2 g COD/g MLSS/d in the AT_1_ of the step-aerating MBR under the imposed operating conditions of DO = 2.5 ± 0.1 mg/L contributed essentially in postponing membrane fouling for more than 3 months (Figure 3—Stage 1), a result further discussed in the following paragraphs. In contrast, the decrease of DO at 1.2 ± 0.5 mg/L resulted in an exponential increase of the TMP within a month (Figure 3—Stage 2), a result associated with the growth of filaments. Given the second graph of Figure 3, TMP value was increased rapidly and reached 14 kPa only in 47 d at the control MBR. Since both MBR units had similar influent COD, MLSS, DO and total F/M loading, it is concluded that the influence of the step-aeration was crucial for the decrease of TMP and the membrane fouling reduction for the step-aerating MBR.

The moderate population of filamentous microorganisms may reduce membrane fouling in MBRs forming sludge of high porosity and low adhesion on the membrane surface [23,24]. The step aeration imposed in the MBR unit by dividing the aeration tank into two sub-chambers, AT_1_ and AT_2_, as well as the imposed F/M and DO values during Stage 1, contributed effectively to the growth of filamentous population in moderate concentrations, corresponding to filamentous index 1.5 < FI < 3 (Figure 4), achieving one of the objectives of this work, to utilize the filamentous bacteria for membrane fouling reduction. Comparing the results of Figure 3 and Figure 4, it is indeed concluded that the control of filamentous microorganisms at moderate concentrations had, as a result, to maintain TMP at really low values, smaller than 2 kPa.

A low total HRT of 18 h and/or 5 h in a wastewater treatment process may cause an increase on SMP and growth of filamentous bacteria according to Fallah et al. [32] and Meng et al. [26] respectively. On the other hand, too high HRT also may lead to the accumulation of foulants [26,32]. For these reasons the total HRT at the current study was adjusted at 28 h (Table 1) aiming not to affect negatively the SMP and filamentous population. The reasoning behind the controlled growth of filamentous bacteria is described as follows. The hydraulic retention time (HRT) was maintained as low as 1.6 h in the AT_1_, while a high amount of food was supplied (high F/M = 0.45–0.85 g COD/g MLSS/d). In this way, filamentous bacteria were increased easily because of high F/M loading, while biomass activity was decreased because of low HRT. Following, in the AT_2_, the HRT was increased slightly at 4.8 h whereas the food supply decreased to a minimum (F/M = 0.02–0.04 g COD/g MLSS/d). Therefore, the growing tendency of filamentous bacteria was prevented, and their growth was brought under control, as a result of low F/M loading. Furthermore, microorganisms were given time to digest their food and increase the biomass activity, as the HRT was increased marginally. In more detail, the AT_1_ chamber constituted the critical step, where high rate bio-absorption of organic matter by bacteria and protozoa took place, having, as a result, the restricted bio-absorption (feeding) of the filamentous bacteria that are known for presenting a lower rate of bio-absorption [47]. Therefore, the growth rate of filamentous bacteria was suppressed to the level of 1.5 < FI < 3.

As a result of the imposed operating conditions at the step-aerating MBR and the controlled filamentous growth, the MBR operation was extended for a long period without the need for intermediate chemical cleaning of the used membrane along with less consumed energy, since the pumping through the membrane was carried out with very low TMP. Therefore it could be concluded that the proposed solution of the step-aerating ΜΒR reduces significantly the operating cost of MBR units in general, by minimizing the frequency of chemical cleaning and the frequency of replacement of the fouled membrane as well as by reducing the energy consumed during the filtration. These benefits could be calculated in future research work with a comparable cost analysis regarding the operation of a step-aerating MBR and a conventional MBR comprised of an aeration tank and a membrane tank.

Going back to the operating conditions, low DO concentrations or DO deficiency increase filamentous population in wastewater treatment processes [27,28,29,30]. In this study, the decrease of DO concentration at 1.2 ± 0.5 mg/L during Stage 2, verified the literature and led to a sharp increase of the filamentous population to FI = 5. DO reduction in the AT_1_ chamber reduced the bio-absorption rate of organic matter by bacteria and protozoa, which favored the parallel bio-absorption of organic matter by the filaments and the increase of their population. Following, the high growth of FI resulted in the aggravation of the membrane fouling (Figure 3), thus verifying other research work [24] and completing other research work [37]. In conclusion, the effect of the high filamentous bacteria population on the membrane-fouling rate was not negligible.

According to Figure 5a–c of optical microscopy, it could be concluded that when filamentous bacteria ranged at 1.5 < FI < 3, they helped to aggregate the sludge flocs and colloids in the mixed liquor acting as a polyelectrolyte, decreasing by this way the TMP and membrane fouling. On the other hand, when filamentous bacteria ranged at FI = 5 (abundance of filaments) (Figure 5d), they intensified the dispersion of sludge flocs in the mixed liquor resulting in an inability to bind and aggregate colloids, launching exponentially the TMP and membrane fouling respectively. This conclusion of whether or not the colloidal components were bound on the sludge flocs was particularly important given that the population of colloids plays a decisive and very important role in membrane fouling [13].

According to Figure 6a,b the greatest population of filamentous microorganisms were *Type 0092* filaments. *Type 0092* filaments are mostly straight, irregularly curved or bent, composed of rectangular cells [27,48] and do not present attached growth or a sheath. These bacteria are often found in environments with low food to microorganism (F/M) ratio (0.02–0.2 g BOD/g MLSS/d ≈ 0.04–0.4 g COD/g MLSS/d) and long SRT (10–40 days). *Type 0092* prospers in conditions with very similar food sources to *M. parvicella* but warmer weather conditions. In many treatment plants, the disappearance of *M. parvicella* in April/May is followed by an increase of *Type 0092,* as both prefer the same substrate. *Type 0092* filaments have been found when the temperature of wastewater is above 15 °C [27]. In this case, the growth of *Type 0092* filaments could be attributed to the low F/M (0.13 ± 0.04 g COD/g MLSS/d) ratio in the total MBR unit, in combination with the high SRT equal to 30 ± 5 d. Moreover, the warm temperature (20 ± 3 °C) contributed to their growth too.

According to Figure 6c,d, *Thiothrix* spp. filaments were, also, detected. *Thiothrix* spp. filaments are Gram-negative, Neisser negative (with some Neisser positive granules), straight or smoothly curved, with rectangular cells having clear septa without indentations and attached growth. A heavy sheath is possibly apparent and easy to identify because of its large size. *Thiothrix* spp. are found in environments with low DO [27]. Therefore, the *Thiothrix* filaments grew in this MBR unit because of the DO decrease in concentration smaller than 2 mg/L during Stage 2 of the MBR operation (Figure 2).

Finally, Figure 6e,f illustrate that the filaments *Microthrix parvicella* were, also, present in the mixed liquor. *Microthrix parvicella* are Gram-positive, Neisser negative, irregularly coiled filaments within the floc or in loose “patches”, free in the bulk solution and they do not present any attached growth. *M. parvicella* are usually present in colder environments at temperatures as low as 7 °C in pure cultures as well as in full-scale plants [49], while high temperature leads to their decrease [50]. These filaments are usually found in environments where the food to microorganism (F/M) ratio is low (0.05–0.2 g BOD/g MLSS/d = 0.1–0.4 g COD/g MLSS/d) and SRT is not too short (>8–10 days). It has also been found that the F/M ratio was negatively correlated with the number of *M. parvicella* gene copies [50]. The growth of *M. parvicella,* in this MBR unit may be attributed to the same factors as *Type 0092* filaments, i.e., the low F/M ratio of 0.13 ± 0.04 g COD/g MLSS/d and the high SRT of 30 ± 5 d. However, their growth was not extensive because of the warm temperatures of 20 ± 3 °C in the MBR unit.

According to some research works [25,26], a high Food/Microorganisms (F/M) ratio increases the SMP concentration, resulting in a decrease in sludge filterability. In this study, despite the high F/M loading in the AT_1_ (0.65 ± 0.2 g COD/g MLSS/d), the soluble microbial products (SMP) were maintained at quite low concentrations, a result that was facilitated by the favorable conditions in the AT_2_ of F/M = 0.03 ± 0.01 g COD/g MLSS/d, in conjunction with the high hydraulic retention time in this tank, of 17 h. SMP are gradually deposited and aggregated in the membrane pores causing irreversible membrane fouling [16], therefore the quite low concentrations of SMP contributed additionally to the maintenance of ΤΜP at low prices and therefore to the prolonged fouling reduction.

Moreover, it is concluded that the SMP were not affected by the decrease of DO in the AT_1_. SMP were neither affected by the increase of filamentous microorganisms from 2 to 5, as it is presented clearly in Figure 7c, complementing the work of other researchers [38], according to which EPS in the form of proteins and polysaccharides gradually decreased during sludge bulking, while no information was provided regarding SMP. Furthermore, these results verified an older research work [37], according to which filamentous bacteria density had no significant effect on SMP. In conclusion, comparing the results of Figure 7a,b with Figure 3, it appears that the TMP increase during Stage 2 was not attributed to a corresponding increase in SMP but only to the uncontrolled growth of filamentous bacteria.

## 5. Conclusions

According to this research work, it was found that the optimal population of filamentous bacteria of 1.5 < FI < 3 that minimized membrane fouling, was achieved by the imposed conditions in the first aerated bioreactor AT_1_, where F/M_AT1_ ≤ 0.65 ± 0.2 g COD/g MLSS/d at a temperature of 20 ± 3 °C, based on the typical concentration of dissolved oxygen in the wastewater treatment plants of 2.5 ± 0.1 mg/L and having a low HRT of 1.6 h. Filamentous bacteria aggregated the sludge flocs and colloids in the mixed liquor, keeping TMP at really low values, smaller than 2 kPa, for a great period (3 months), while at the control MBR the TMP was gradually increased reaching 14 kPa in just 1.5 months. After that, the effects of a decrease of the DO concentration in the AT_1_ at 1.2 ± 0.5 mg/L at the step-aerating MBR were checked, which led to a growth of the filamentous population to FI = 5. The high growth of the filamentous population resulted in the dispersion of sludge flocs and colloids in the mixed liquor presenting an inability to connect and aggregate, launching exponentially the TMP and membrane fouling respectively. On the last day of total membrane fouling, the types of filamentous microorganisms were identified. The main type of filament detected was *Type 0092*, while *Thiothrix* spp. and *Microthrix parvicella* were also detected. Furthermore, during the entire MBR operation period, the SMP were maintained at quite low concentrations, with SMP carbohydrates < 15 mg/L and proteins < 20 mg/L. The quite low concentrations of SMP contributed additionally to the maintenance of ΤΜP at low prices during Stage 1. SMP were not affected by the decrease of DO in the AT_1_, nor by the increase of FI from 2 to 5 during Stage 2. Finally, wastewater treatment performance was excellent.

## Figures and Tables

**Figure 1 membranes-11-00553-f001:**
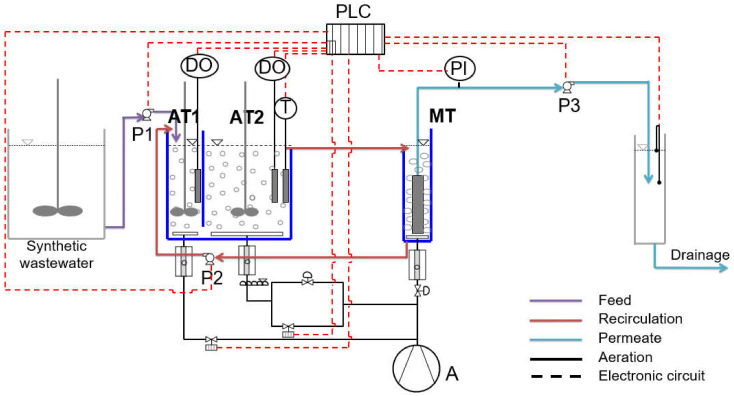
Flowchart of the step aerating MBR with two in-series aerated tanks, the AT_1_ (5 L) and the AT_2_ (15 L), followed by the membrane tank MT (5 L).

**Figure 2 membranes-11-00553-f002:**
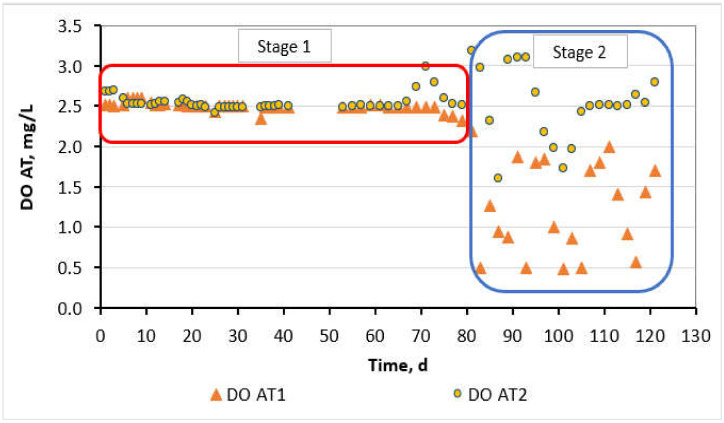
DO concentration in the AT_1_ and AT_2_ chambers at the step-aerating MBR as a function of the operation time.

**Figure 3 membranes-11-00553-f003:**
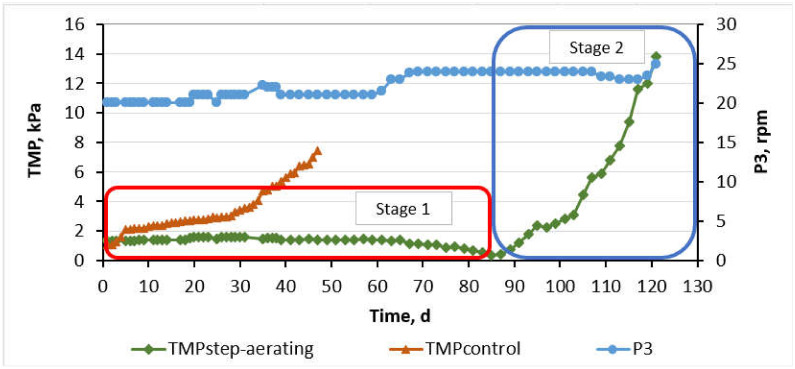
TMP graphs for the step-aerating MBR and control MBR, and peristaltic pump P_3_ rotations versus the MBR operation time.

**Figure 4 membranes-11-00553-f004:**
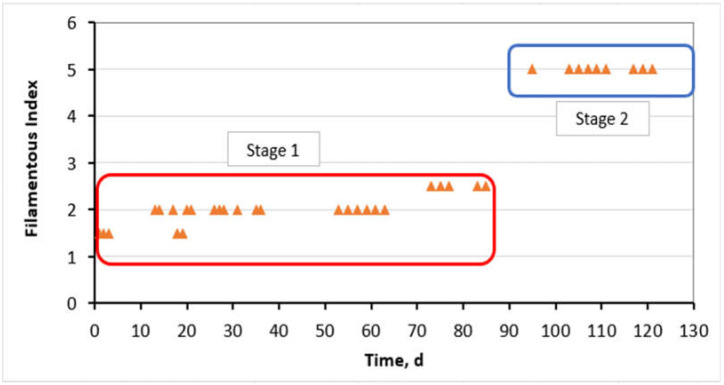
Filamentous index versus the MBR operation time.

**Figure 5 membranes-11-00553-f005:**
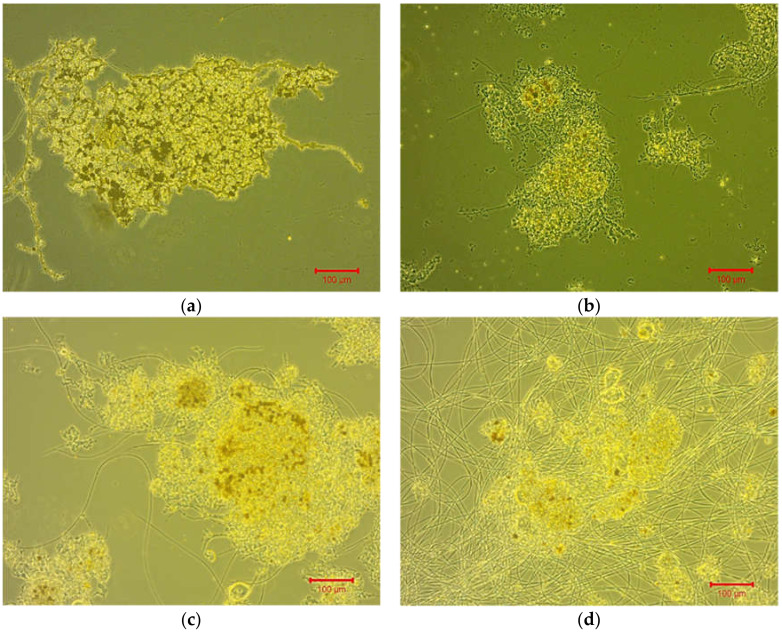
Optical microscopy images for the activated sludge mixed liquor’s samples of the step-aerating MBR, where (**a**) FI = 1.5 on day 18, (**b**) FI = 2 on day 27, (**c**) FI = 2.5 on day 75 and (**d**) FI = 5 on day 121.

**Figure 6 membranes-11-00553-f006:**
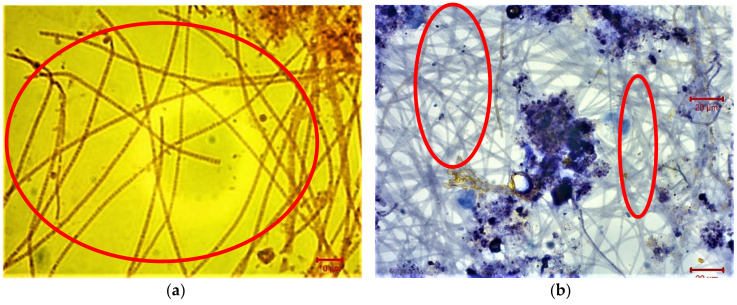
Optical microscopy images of (**a**) Gram-negative and (**b**) Neisser positive filaments named as *Type 0092*, (**c**,**d**) Neisser negative filaments named as *Thiothrix* and (**e**,**f**) Neisser negative filaments named as *Microthrix parvicella,* at the 121st day of the maximum membrane fouling.

**Figure 7 membranes-11-00553-f007:**
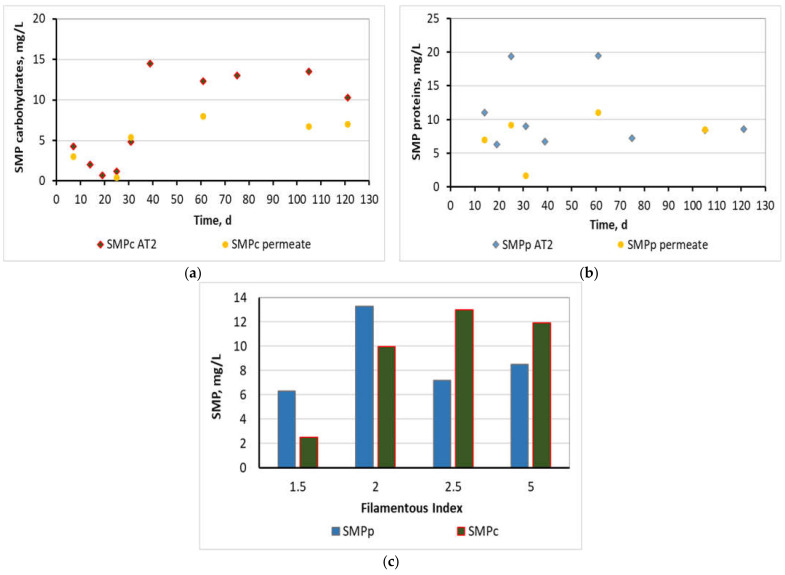
(**a**) SMP carbohydrates and (**b**) SMP proteins in the mixed liquor of the AT_2_ and in the permeate as a function of the MBR operation time and (**c**) correlation of SMP concentration with Filamentous Index.

**Table 1 membranes-11-00553-t001:** Operating parameters of the step-aerating MBR and control MBR units.

Operating Parameters	Step-Aerating MBR—Stage 1	Step-Aerating MBR—Stage 2	Control MBR
Working time, d	0–81	82–121	47
DO_AT1_, mg/L	2.5 ± 0.1	1.2 ± 0.5	2.5 ± 0.1 (at the unique AT)
DO_AT2_, mg/L	2.5 ± 0.1	2.5 ± 0.4	-
F/M_AT1_, g COD/g MLSS/d	0.65 ± 0.20		0.14 ± 0.03 (at the unique AT)
F/M_AT2_, g COD/g MLSS/d	0.03 ± 0.01		-
F/M_tot_, g COD/g MLSS/d	0.13 ± 0.04		0.12 ± 0.03
Recirculation rate/Feed rate	2.4:1		2.4:1
HRT_AT1_, h, based on Q_in_ + Q_r_ (HRT_AT1_, h, based on Q_in_)	1.6 (5.5)		6.5 (22) (at the unique AT)
HRT_AT2_, h, based on Q_in_ + Q_r_ (HRT_AT2_, h, based on Q_in_)	4.8 (17)		-
HRT_tot_, h	28		28
SRT, d	30 ± 5 d		30 ± 5 d
MLSS, mg/L	6000 ± 1500		6000 ± 1000
Temperature, °C	20 ± 3 °C		20 ± 3 °C

**Table 2 membranes-11-00553-t002:** Wastewater treatment efficiency in the step-aerating MBR.

Parameters	Concentration (mg/L)	
	Influent	Effluent
COD	890 ± 109	15 ± 5.7
NO_3_-N	0.37 ± 0.21	37 ± 11
NH_4_-N	31 ± 7.1	0.06 ± 0.09
TN	63 ± 7.7	48 ± 12

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
