# Peer review of "Membrane Fouling Controlled by Adjustment of Biological Treatment Parameters in Step-Aerating MBR"

_membranes, 2021, doi:10.3390/membranes11080553_

Round 1
Reviewer 1 Report
This study investigates the membrane fouling reduction through the adjustment of operating parameters for sludge morphology modification. The writing is clear. Readers in the fields of biological wastewater treatment will be interested in this work. It is recommended for minor revision.
Some suggestions are listed below:
- The main hypothesis and research approach were not sufficiently discussed. e.g. Given that the impacts of those operating factors are highly interrelated, how to address the complexity.
- To what extent can the results further improves the efficiency of bio-process control. e.g. energy saving. Can it be quantified?
- Two relevant references about biofilm fouling and shear are recommenced: Quantitative characterization and analysis of granule transformations: Role of intermittent gas sparging in a super high-rate anaerobic system; A super high-rate sulfidogenic system for saline sewage treatment
- Filaments were discussed and it would be great if the data of their abundance was available
Reviewer 2 Report
The manuscript was well written and data are clearly presented and discussed. Several comments are listed below:
- Line 161: “…effluent (Qeff) flowrates”, Q refers to flowrate, please make a change for others.
- Figure 2: removing the second y-axis as it is the same as the first y-axis. The title of y-axis shall be DO (mg/L). Label “Stage 1” and “Stage 2” in the figure.
- Figure 3: the flowrate (or membrane flux) shall be presented in the second y-axis instead of pump speed (rpm). Both figures 3 and 4 shall be combined.
- The authors shall clarify if the cleaned membrane was used in Stage 2 (i.e., if replacing the membrane after stage 1?)
- The authors shall provide a summary table for operation conditions of control MBR and step-aerating MBR in materials and methods part.
- Any filamentous index data in control MBR?
- The identification of filamentous bacteria using microscope and dying method cannot provide the spices information (needs DNA sequencing). Section 3.4 needs to be carefully revised and combined with section 3.3.
- The conclusions should be shortened to summarize the key findings.
Reviewer 3 Report
Manuscript entitled “Membrane fouling controlled by adjustment of biological treatment parameters in step-aerating MBR” submitted by Dimitra C. Banti, Manassis Mitrakas, Petros Samaras, can be accepted for publication in Membranes Journal, after a minor revision.
Here is a list of my specific comments:
- Page 1, Keywords: The number of keywords is too high and should be reduced.
- Page 5, 3. Results: This section is too long. Here should be only presented the experimental results. Therefore, all observations and discussions should be moved in the next section.
- Page 11, 4. Discussion: This section is too brief and should be detailed. All the experimental results presented in the previous section should be clearly and detailed discussed here.
- Page 12, 5. Conclusions: This section is quite too long and should be systematized. Pay attention on the most important experimental results and findings to highlight the importance of this study.
Reviewer 4 Report
The work presented and the results obtained are interesting. Please see the following comments: 1. The manuscript should be improved by significantly comparing the obtained results with those presented in the available literature.2. References should be supplemented with the latest literature reports. The reference list contains only 12 studies published in the last 5 years.
3. Line 57: Repeated explanation of the abbreviation: 'dissolved oxygen (DO)'.
4. Although the abbreviation for EPS is well known, it should be explained.
